# Counterfactual Analysis for Digital Histopathology Slides Using Human Interpretable Features

**Hakim Benkirane**[1,2]                               HAKIM.BENKIRANE@CENTRALESUPELEC.FR
[1] *Laboratory MICS, CentraleSupelec, Paris-Saclay University, Gif-sur-Yvette, France*
[2] *INSERM U1018, CESP, Gustave Roussy, Villejuif, France*

**Maria Vakalopoulou**[1]                          MARIA.VAKALOPOULOU@CENTRALESUPELEC.FR
**Stefan Michiels**[2,3]                              STEFAN.MICHIELS@GUSTAVEROUSSY.FR
[3] *Bureau de Biostatistique et d'Épidémiologie, Gustave Roussy, Paris-Saclay University*
**Paul-Henry Cournède**[1]                        PAUL-HENRY.COURNEDE@CENTRALESUPELEC.FR
**William Lotter**[4,5]                                  LOTTERB@DS.DFCI.HARVARD.EDU
[4] *Dana-Farber Cancer Institute, Department of Data Science, Boston, Massachusetts*
[5] *Harvard Medical School, Boston, Massachusetts*

**Editors:** Under Review for MIDL 2024

## Abstract

Recent advancements in deep learning techniques have greatly improved the precision and efficiency of computational pathology processes, facilitating diagnosis, outcome forecasting, and identification of genetic markers and disease progression. However, a significant challenge hindering the integration of these computational tools into clinical practice is the lack of interpretability of their results. In this paper, we propose a novel method for counterfactual analysis on histopathology slides to provide clear and understandable explanations based on human interpretable features for predictive tasks at the slide level. Our method addresses the challenge of generating interpretable explanations for high-dimensional tabular data in a computationally efficient manner, outperforming state-of-the-art methods by generating explanations approximately 10 times faster. This advancement holds promise for enhancing the adoption and effectiveness of deep learning models in clinical settings, ultimately improving patient care and outcomes.

**Keywords:** Histopathology, Whole slide Imaging, Counterfactual, Interpretability

## 1. Introduction

In recent years, deep learning has significantly advanced computational pathology, achieving breakthroughs in disease diagnosis, outcome prediction, and genetic marker identification from histopathology images (Lu et al., 2021; Shao et al., 2021; Benkirane et al., 2022). However, the interpretability of these models remains a significant hurdle, despite their accuracy, limiting their adoption in clinical practice where transparent insights are crucial.

To address this gap, some studies have explored leveraging human-interpretable features (HIFs) to characterize histopathology slides, translating complex tissue structures into easily understandable tabular data (Diao et al., 2021). While this approach offers comprehensive descriptions of whole slide images (WSIs), models using HIFs lack the predictive power of image-based deep learning models. Thus, an ideal approach would combine the impressive performance of image-based models while offering the interpretability afforded by HIFs.

Counterfactual analysis, which identifies minimal changes to input data altering model predictions, presents a promising strategy for interpretability (Pawelczyk et al., 2020; Yang et al., 2022). However, applying this analysis to high-dimensional tabular data, such as the ones describing gigapixel-sized WSIs, poses unique challenges. Existing methods, designed for fewer features, often suffer from computational inefficiency when searching for counterfactual examples (Mothilal et al., 2020; Yang et al., 2022). In response, our paper introduces a novel approach tailored for high-dimensional tabular data, aiming to efficiently generate interpretable explanations. By bridging the interpretability gap, our method facilitates the integration of deep learning models into clinical practice, offering clear and actionable insights into model decisions.

## 2. Methods

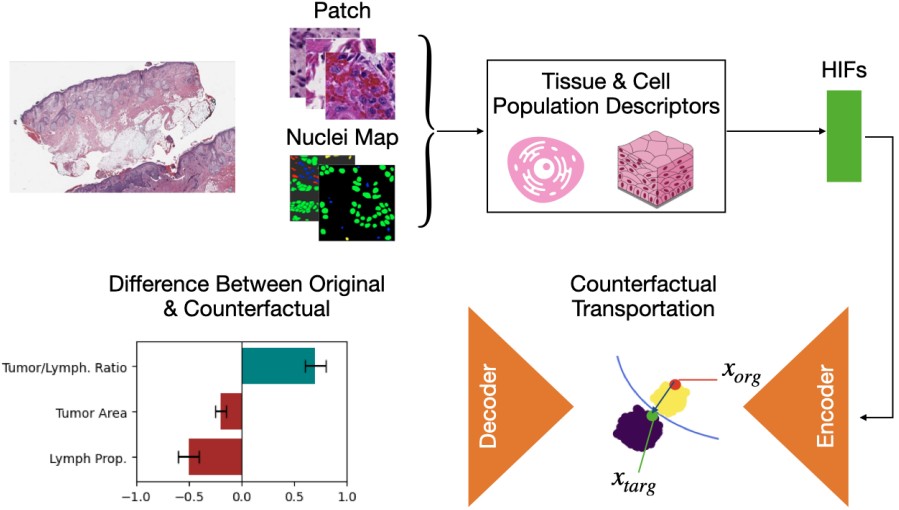

Figure 1: **Overview of the Method:** Human Interpretable Features are fed to train the GMM normalizing flow model to separate classes. To generate a counterfactual example, the sample point is projected to the boundary between the gaussians with a tolerance threshold.

**Extraction of Human Interpretable Features.** In our study, we use a method from (Diao et al., 2021) to extract human-interpretable features from WSIs, including area measurements of tissue regions, proportional analysis of cell and tissue types, and cell clustering properties, which offer insights into cellular makeup and potential abnormalities.

**GMM-CeFlow: A GMM-based normalizing flow for efficient counterfactual generation in high-dimensional space.** In our approach, we develop GMM-CeFlow (Gaussian Mixture Model Counterfactual Flows) to generate counterfactual explanations. It extends the idea from Dung Duong et al. (2023) by using a GMM-based latent space to allow for a a more efficient counterfactual transportation with a tractable formula. We employ

a normalizing flows-based framework, utilizing RealNVP (Dinh et al., 2016), to transform data into a latent space conducive to manipulating class distributions. This latent space consists of a mixture of Gaussian distributions, each corresponding to a class similarly to Izmailov et al. (2020), enabling controlled manipulation for counterfactual generation. We enforce sparsity constraints through a L1-loss on the means of these Gaussians to minimize the number of features changed during counterfactual generation. To generate a counterfactual, we encode the original instance $x_i$ to its latent representation $z_{\mathrm{org}}$. We then find the optimal counterfactual representation $z_{\mathrm{targ}}$ in the target class by projecting $z_{\mathrm{org}}$ onto the decision boundary between the original and target classes, minimizing the required perturbation. Since the decision boundary is represented as a quadric hypersurface, we utilize the approach introduced in (Van Hoorebeeck and Absil, 2022). We thus gain significant computation efficiency compared to methods that requires the search for counterfactual in the whole space. An overview of our proposed method is presented in Fig. 1.

## 3. Results & Discussion

We evaluate our method on a public breast cancer dataset (TCGA-BRCA, $n = 1187$) for a task of predicting molecular subtypes PAM50 to showcase the model's ability to handle multi-task counterfactual generation. To compare our method against other state-of-the-art approaches, we use the same evaluation metrics as in (Bodria et al., 2022).

Table 1: Assessment of counterfactual methods using various criteria: a) Proximity between counterfactual and original sample (PROX), b) Average feature changes in counterfactuals (CNT), c) Implausibility of counterfactual explanations (IMP), d) Counterfactual success in class change (SR), e) Average counterfactual generation time across 10 runs (min) (TIM).

| Model | PROX | CNT | IMP | SR | TIM |
|---|---|---|---|---|---|
| MACE (Yang et al., 2022) | $0.79 \pm 1.00$ | $10.54 \pm 2.89$ | $0.67 \pm 0.45$ | 91% | $103, 9 \pm 7, 0$ |
| DICE (Mothilal et al., 2020) | $0.81 \pm 0.54$ | $\mathbf{9.27 \pm 2.51}$ | $0.42 \pm 0.38$ | 87% | $91, 2 \pm 10, 0$ |
| T-LACE (Bodria et al., 2022) | $\mathbf{0.90 \pm 0.30}$ | $9.31 \pm 3.12$ | $0.44 \pm 0.12$ | 97% | $82, 4 \pm 9, 7$ |
| Our proposed | $0.85 \pm 0.44$ | $9.74 \pm 3.14$ | $\mathbf{0.39 \pm 0.22}$ | $\mathbf{99\%}$ | $\mathbf{14, 7 \pm 3, 5}$ |

As demonstrated in Table 1, our method achieves comparable performance to other state-of-the-art methods, yet notably accomplishes this within significantly reduced computation time. This efficiency stems from our use of tractable, easily computed formulas, in contrast to the greedy search approach employed by other methods for counterfactual explanations. However, it's important to note that while our results affirm the effectiveness of our method, utilizing an estimation of the classifier through the GMM latent space, trained with the classifier, rather than the original black-box classifier, poses a challenge for generalizability. Further validation across multiple datasets is necessary to ensure the model's robustness. Additionally, while this study exclusively employs interpretable histopathology features for clarity in counterfactual examples, future research should explore analyses in the original image space.

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
