# OpenReview forum: "Counterfactual Analysis for Digital Histopathology Slides Using Human Interpretable Features"
_MIDL.io/2024/Short_Papers — MIDL 2024 Short Papers_

### Official Review · Reviewer_krLK · 2024-04-23

**Confidence:** 3
**Final Rating:** 3.5

**Review:**

Summary
-------

The paper proposes a counterfactual explanation method for histopathology grading. The images are first characterised with a vector of human-interpretable features (HIFs), and then the counterfactuals are provided on the tabular feature level.


Strengths
---------

- The authors creatively combine existing work on human-interpretable feature extraction of whole slide images and tabular counterfactual generation.
- The proposed report of tabular counterfactuals (last panel in Fig. 1) could be appealing in a clinical workflow.


Weaknesses
----------

- The authors state that their approach should "combine the impressive performance of image-based models while offering the interpretability afforded by HIFs". It would have been useful to report the predictive performance in the results section. Furthermore, it would be great if the authors could elaborate how their approach actually exploits the superior performance of image-based models. From what I understand, it seems that the images are not used except for feature extraction, but maybe I misunderstood.

---

### Decision · Program_Chairs · 2024-04-26

Accept